# Predictors of severity and development of critical illness of Egyptian COVID-19 patients: A multicenter study

**Dalia Omran** [1]*, **Mohamed Al Soda**[2], **Eshak Bahbah**[3], **Gamal Esmat**[1], **Hend Shousha**[1], **Ahmed Elgebaly**[4], **Muhammad Abdel Ghaffar**[5], **Mohamed Alsheikh**[6], **Enass El Sayed**[7], **Shimaa Afify**[8], **Samah Abdel Hafez**[5], **Khaled Elkelany**[9], **Ayman Eltayar**[10], **Omnia Ali**[11], **Lamiaa Kamal**[12], **Ahmed Heiba** [5,13]

1 Department of Endemic Medicine and Hepatology, Faculty of Medicine, Cairo University, Cairo, Egypt,
2 General Organization for Teaching Hospitals and Institutes, Cairo, Egypt, 3 Faculty of Medicine, Al-Azhar University, Damietta, Egypt, 4 Faculty of Medicine, Al-Azhar University, Cairo, Egypt, 5 Gastroenterology & Infectious Diseases Department, Ahmed Maher Teaching Hospital, Cairo, Egypt, 6 Chest Unit, Al Matareyah Teaching Hospital, Cairo, Egypt, 7 Nephrology Department, Ahmed Maher Teaching Hospital, Cairo, Egypt, 8 Gastroenterology Department, National Hepatology and Tropical Medicine Research Institute, Cairo, Egypt, 9 Pediatric Department, Shebin Elkom Teaching Hospital, Shebin Elkom, Egypt, 10 Intensive care Department, Damanhour Teaching Hospital, Damanhour, Egypt, 11 Clinical and Chemical Pathology Department, Ahmed Maher Teaching Hospital, Cairo, Egypt, 12 Clinical and Chemical Pathology Department, Elsahel Teaching Hospital, Cairo, Egypt, 13 Internal Medicine Department, National Research Centre, Cairo, Egypt

* daliaomran@kasralainy.edu.eg

## Abstract

### Objectives

We conducted the present multicenter, retrospective study to assess the epidemiological, clinical, laboratory, and radiological characteristics associated with critical illness among patients with COVID-19 from Egypt.

### Methods

The present study was a multicenter, retrospective study that retrieved the data of all Egyptian cases with confirmed COVID-19 admitted to hospitals affiliated to the General Organization for Teaching Hospitals and Institutes (GOTHI) through the period from March to July 2020. The diagnosis of COVID-19 was based on a positive reverse transcription-polymerase chain reaction (RT-PCR) laboratory test.

### Results

This retrospective study included 2724 COVID-19 patients, of whom 423 (15.52%) were critically ill. Approximately 45.86% of the critical group aged above 60 years, compared to 39.59% in the non-critical group (p = 0.016). Multivariate analysis showed that many factors were predictors of critically illness, including age >60 years (OR = 1.30, 95% CI [1.05, 1.61], p = 0.014), low oxygen saturation (OR = 0.93, 95% CI [0.91, 0.95], p<0.001), low Glasgow coma scale (OR = 0.75, 95% CI [0.67, 0.84], p<0.001), diabetes (OR = 1.62, 95% CI [1.26,

**Data Availability Statement:** The data underlying the results presented in the study are available from the database of the Egyptian general organization of teaching hospitals and institutes

(GOTHI) but cannot be shared publicly due to local restrictions. Interested researchers may contact Dr. Ahmed Abdel Salam (ahmedabdelsalam@sevoclin. com; +201551475160) or the corresponding author, Dr. Dalia Omran.

**Funding:** This research work received no fund.

**Competing interests:** All authors declare no conflict or competing interest.

2.08], p<0.001), cancer (OR = 2.47, 95% CI [1.41, 4.35], p = 0.002), and serum ferritin (OR = 1.004, 95% CI [1.0003, 1.008], p = 0.031).

## Conclusion

In the present report, we demonstrated that many factors are associated with COVID-19 critical illness, including older age groups, fatigue, elevated temperature, increased pulse, lower oxygen saturation, the preexistence of diabetes, malignancies, cardiovascular disease, renal diseases, and pulmonary disease. Moreover, elevated serum levels of ALT, AST, and ferritin are associated with worse outcomes. Further studies are required to identify independent predictors of mortality for patients with COVID-19.

## 1. Introduction

Since the discovery of the initial cluster of cases in Wuhan, China, the coronavirus disease 2019 (COVID-19) outbreak has emerged as one the greatest health threat and was declared as a pandemic by the World Health Organization (WHO) on March 11, 2020 [1, 2]. By the end of October 2020, the COVID-19 outbreak has affected more than 200 countries in various regions of the world; according to the WHO situation reports, the number of COVID-19 confirmed cases reached more than 41,571,000 cases and 1,135,000 related deaths worldwide [3]. In the past few months, the COVID-19 pandemic landscape has changed with a dramatic shift in the apex of the number of confirmed cases from China to the United States (US), Europe, and Eastern Mediterranean Region (EMR) [4, 5]. In the EMR, the number of confirmed cases is approaching three million cases by October 2020, with a fatality rate of 2.6% [6]. While the causative agents and mode of transmission remained ambiguous during the early day of outbreaks, it is now clear that COVID-19 is a highly contagious disease that is caused by Severe acute respiratory syndrome coronavirus 2 (SARS-CoV-2), which is transmitted mainly by respiratory droplets [7, 8]. Besides, the current body of evidence demonstrates that the infection can be transmitted by both symptomatic and asymptomatic cases leading to the dramatic spread of the outbreak within the community [9].

The presentation and outcomes of COVID-19 vary substantially, with clinical features ranging from asymptomatic/mild symptoms to fatal respiratory distress and multi-organ failure [10, 11]. Previous reports demonstrated that symptomatic patients usually present with flu-like symptoms that resolve completely by the end of the disease course [12]; nonetheless, a subset of the patients can present with severe forms of the disease, including severe pneumonia, acute respiratory distress syndrome (ARDS), sepsis, thromboembolic manifestations, acute myopericarditis, septic shock, multi-organ failure, and eventually death [13–16]. While the exact pathogenic mechanisms underlying the development of severe forms have not been fully elucidated yet, recent experiments highlighted that patients with severe COVID-19 exhibit exaggerated inflammatory process and release of inflammatory cytokines, which in rerun induce a cytokine storm and organ damage [17].

Recent studies confirmed that the COVID-19 outbreak does not exert an equal toll amongst the different regions affected, with wide variation in the proportion of patients with severe disease and mortality [18]. While the adequacy of healthcare services may play a role in such inconsistencies [19], multicenter reports highlighted that patient-specific factors are major determinants of the presentation and outcomes of COVID-19. Old age, male gender, comorbidities, and immunocompromised status were reported to be associated with severe

presentation and poor disease outcomes [20, 21]; besides, predisposing genetic factors may play a role in determining an individual's susceptibility to infection and disease course as well [22, 23].

Being a frequent destination of tourism, international traffic and a densely populated country, Egypt observed an increasing spread of COVID-19 since March 5; the official figures demonstrated the number of confirmed cases was nearly 108,000 by the end of October 2020 with a fatality rate of 5.8% [24]. Besides, the mortality rate among hospitalized patients with COVID-19 in Egypt was reported to be 6.7% [25], which rises to 40% in severe and critically-ill cases [26]. Such a fatality rate is notably higher than in other countries in the EMR, which can be attributed to the potential role of patient-specific characteristics. Thus, we conducted the present retrospective study to assess the epidemiological, clinical, laboratory, and radiological characteristics associated with critical illness development among patients with COVID-19 from Egypt.

## 2. Materials and methods

The present study was initiated after obtaining the protocol approval from the General Organization for Teaching Hospitals and Institutes (GOTHI) responsible ethics committees in Egypt (IRB:HAM00122). The preparation of the present manuscript runs in compliance with the recommendations of the STROBE statement [27].

### 2.1. Study design and population

The present study was a multicenter, retrospective study that retrieved the data of all Egyptian cases with confirmed COVID-19 admitted to hospitals affiliated to the GOTHI through the period from March to July 2020 The diagnosis of COVID-19 was based on a positive reverse transcription-polymerase chain reaction (RT-PCR) laboratory test.

### 2.2. Data collection and operational definitions

We retrieved epidemiological, clinical, laboratory, and radiological data of all eligible patients. The epidemiological characteristics included age, gender, and smoking, while the clinical data included the COVID-19 symptoms and the associated comorbidities. The laboratory and radiological data included complete blood count (CBC) with differential count, C-reactive protein (CRP), serum ferritin, liver function tests, renal function tests, coagulation profile, D-Dimer level, arterial blood gas analysis, and chest imaging. The data were collected at the first day of hospitalization of each patient.

According to Chinese guidelines for the management of COVID-19; Severe COVID-19 was defined as the presence of radiological evidence of more than 50% lung infiltrate plus one of the following: respiratory rate of ≥ 30 breaths per minute; oxygen saturation (SaO2) <94% while breathing ambient air at rest; or ARDS that is defined as arterial oxygen partial pressure (PaO2) to a fraction of inspired oxygen (FiO2) (PaO2: FiO2) of ≤300 mmHg. In addition, critical COVID-19 was defined as respiratory failure requiring ventilator support either invasive or none, septic shock, and/or any organ dysfunction that needs supportive treatment in ICU [28].

### 2.3. Statistical analysis

Data were analyzed using the software Stata 16 for Windows. Categorical variables were summarized by frequency counts and percentages. Continuous variables were represented as means and standard deviations. Comparison between moderate and severe cases to critically-

ill cases was done through univariate analysis as follows: categorical variables were assessed with the Chi-square test, whereas continuous variables were assessed using the Mann-Whitney test. Multivariate analysis was also done to determine factors associated with patients' deterioration. The odds ratio (OR) was calculated by using stepwise logistic regression modeling. Accordingly, the regression equation was used to calculate predicted probabilities of patients' deterioration. Multivariate regression was validated using the Hosmer-Lemeshow goodness-of-fit test.

## 3. Results

### 3.1. Demographic characteristics

This retrospective study included 2724 COVID-19 patients, of whom 423 (15.52%) were critically ill. Approximately 45.86% of the critical group aged above 60 years, compared to 39.59% in the non-critical group (p = 0.016). There was no significant difference between both groups in terms of gender (p = 0.655) and smoking (p = 0.218), Table 1.

### 3.2. COVID-19 symptoms

Regarding the history of contact with infected persons, 25.06% of the critical group's patients had a contact history, compared to 18.47% in the non-critical group (p = 0.002). Our analysis demonstrated no significant difference between both groups in regards to the presence of fever (p = 0.163), cough (p = 0.212), dyspnea (p = 0.390), sore throat (p = 0.283), hemoptysis (p = 0.055), headache (p = 0.298), diarrhea (p = 0.171), nausea (p = 0.703), vomiting (p = 0.646), abdominal pain (p = 0.972), loss of taste (p = 0.151), and loss of smell (p = 0.239). On the other hand, prevalence of fatigue (32.15% vs. 23.73%, p<0.001), anorexia (5.44% vs. 2.96%, p = 0.009), arthralgia (13.71% vs. 6.48%, p<0.001), and myalgia (16.31% vs. 8.52%, p<0.001) was significantly higher in the critical group compared with non-critical group, respectively (Table 1).

### 3.3. Co-morbidities

Patients with critical illness were associated with higher prevalence of diabetes (39.72% vs. 27.86%, p<0.001), cancer (4.49% vs. 1.91%, p = 0.001), hypertension (33.57% vs. 27.03%, p = 0.006), coronary artery diseases (12.53% vs. 6.95%, p<0.001), chronic renal insufficiency (7.09 vs. 2.74%, p<0.001), and asthma (1.42% vs. 0.78%, p = 0.006), Table 1.

### 3.4. CT findings

Interestingly, there was no significant difference (p = 0.566) between critical and non-critical groups in terms of CT findings, including consolidation (3.07% vs. 2.39%), Crazy paving (1.18% vs. 1%), and Ground Glass Opacity (39.95% vs. 44.89%), Table 1.

### 3.5. Vital signs and O$_2$ saturation

There was no significant difference between both groups in terms of systolic and diastolic blood pressure (p = 0.762 and p = 0.577, respectively). On the other hand, the mean value of the pulse (p = 0.006), temperature (p<0.001), and respiratory rate (p<0.001) was higher in critically ill patients compared to the non-critical group. Moreover, the FiO2 was significantly (p = 0.004) higher in the critical group (35.5±33.91) compared to the non-critical group (20.71 ±28.08). In terms of oxygen saturation, critically ill patients were associated with lower O2 saturation (85.73±10.6 vs. 91.04±6.23, p<0.001) and partial pressure of O2 (64.43±26.09 vs 55.22

**Table 1. Demographic, clinical, and radiological characteristics of included patients.**

| Parameters | | Total | Non-critically COVID-19 | Critically COVID-19 | P-value |
|---|---|---|---|---|---|
| | | (n = 2724) | (n = 2301) | (n = 423) | |
| Age group | 0–9 | 14 (0.51%) | 12 (0.47%) | 2 (0.47%) | P = 0.119 |
| | 10–19 | 17 (0.62%) | 16 (0.70%) | 1 (0.24%) | |
| | 20–29 | 108 (3.96%) | 94 (4.09%) | 14 (3.31%) | |
| | 30–39 | 291 (10.68%) | 263 (11.43%) | 28 (6.62%) | |
| | 40–49 | 519 (19.05%) | 444 (19.30%) | 75 (17.73%) | |
| | 50–59 | 670 (24.60%) | 561 (24.38%) | 109 (25.77%) | |
| | 60–69 | 629 (23.09%) | 512 (22.25%) | 117 (27.66%) | |
| | 70–79 | 385 (14.13%) | 323 (14.04%) | 62 (14.66%) | |
| | 80–89 | 86 (3.16%) | 72 (3.13%) | 14 (3.31%) | |
| | 90–99 | 4 (0.15%) | 3 (0.13%) | 1 (0.24%) | |
| | ≥100 | 1 (0.04%) | 1 (0.04%) | 0 (0.00%) | |
| Age > 60 years | No | 1619 (59.43%) | 1390 (60.41%) | 229 (54.14%) | **P = 0.016** |
| | Yes | 1105 (40.57%) | 911 (39.59%) | 194 (45.86%) | |
| Gender | Male | 1396 (51.25%) | 1175 (51.06%) | 221 (52.25%) | 0.655 |
| | Female | 1328 (48.75%) | 1126 (48.94%) | 202 (47.75%) | |
| Smoking | No | 2387 (87.63%) | 2024 (87.96%) | 363 (85.82%) | 0.218 |
| | Yes | 287 (10.54%) | 227 (9.87%) | 60 (14.18%) | |
| Contact History | No | 2193 (80.51%) | 1876 (81.53%) | 317 (74.94%) | **0.002** |
| | Yes | 531 (19.49%) | 425 (18.47%) | 106 (25.06%) | |
| Asymptomatic | No | 2474 (90.82%) | 2091 (90.87%) | 383 (90.54%) | 0.829 |
| | Yes | 250 (9.18%) | 210 (9.13%) | 40 (9.46%) | |
| Fever | No | 792 (29.07%) | 681 (29.60%) | 111 (26.24%) | 0.163 |
| | Yes | 1932 (70.93%) | 1620 (70.40%) | 312 (73.76%) | |
| Cough | No | 1118 (41.04%) | 956 (41.55%) | 162 (38.30%) | 0.212 |
| | Yes | 1606 (58.96%) | 1345 (58.45%) | 261 (61.70%) | |
| Dyspnea | No | 834 (30.62%) | 697 (30.29%) | 137 (32.39%) | 0.390 |
| | Yes | 1890 (69.38%) | 1604 (69.71%) | 286 (67.61%) | |
| Sore throat | No | 1927 (70.74%) | 1637 (71.14%) | 290 (68.56%) | 0.283 |
| | Yes | 797 (29.26%) | 664 (28.86%) | 133 (31.44%) | |
| Hemoptysis | No | 2714 (99.63%) | 2295 (99.74%) | 419 (99.05%) | 0.055 |
| | Yes | 10 (0.37%) | 6 (0.26%) | 4 (0.95%) | |
| Headache | No | 2717 (99.74%) | 2296 (99.78%) | 421 (99.53%) | 0.298 |
| | Yes | 7 (0.26%) | 5 (0.22%) | 2 (0.47%) | |
| Fatigue | No | 2042 (74.96%) | 1755 (76.27%) | 287 (67.85%) | **<0.001** |
| | Yes | 682 (25.04%) | 546 (23.73%) | 136 (32.15%) | |
| Anorexia | No | 2633 (96.66%) | 2233 (97.04%) | 400 (94.56%) | **0.009** |
| | Yes | 91 (3.34%) | 68 (2.96%) | 23 (5.44%) | |
| Diarrhea | No | 2552 (93.69%) | 2162 (93.96%) | 390 (92.20%) | 0.171 |
| | Yes | 172 (6.31%) | 139 (6.04%) | 33 (7.80%) | |
| Nausea | No | 2565 (94.16%) | 2165 (94.09%) | 400 (94.56%) | 0.703 |
| | Yes | 159 (5.84%) | 136 (5.91%) | 23 (5.44%) | |
| Vomiting | No | 2556 (93.83%) | 2157 (93.74%) | 399 (94.33%) | 0.646 |
| | Yes | 168 (6.17%) | 144 (6.26%) | 24 (5.67%) | |
| Abdominal pain | No | 2646 (97.14%) | 2235 (97.13%) | 411 (97.16%) | 0.972 |
| | Yes | 78 (2.86%) | 66 (2.87%) | 12 (2.84%) | |

(*Continued*)

**Table 1.** (Continued)

| Parameters | | Total | Non-critically COVID-19 | Critically COVID-19 | P-value |
|---|---|---|---|---|---|
| | | (n = 2724) | (n = 2301) | (n = 423) | |
| Arthralgia | No | 2517 (92.40%) | 2152 (93.52%) | 365 (86.29%) | <**0.001** |
| | Yes | 207 (7.60%) | 149 (6.48%) | 58 (13.71%) | |
| Myalgia | No | 2459 (90.27%) | 2105 (91.48%) | 354 (83.69%) | <**0.001** |
| | Yes | 265 (9.73%) | 196 (8.52%) | 69 (16.31%) | |
| Loss of taste | No | 2597 (95.34%) | 2188 (95.09%) | 409 (96.69%) | 0.151 |
| | Yes | 127 (4.66%) | 113 (4.91%) | 14 (3.31%) | |
| Loss of smell | No | 2634 (96.70%) | 2221 (96.52%) | 413 (97.64%) | 0.239 |
| | Yes | 92 (3.38%) | 82 (3.56%) | 10 (2.36%) | |
| Number of symptoms | <7 | 26 (0.95%) | 21 (0.91%) | 5 (1.18%) | 0.586 |
| | ≥7 | 2698 (99.05%) | 2280 (99.09%) | 418 (98.82%) | |
| Diabetes | No | 1915 (70.30%) | 1660 (72.14%) | 255 (60.28%) | <**0.001** |
| | Yes | 809 (29.70%) | 641 (27.86%) | 168 (39.72%) | |
| Cancer | No | 2661 (97.69%) | 2257 (98.09%) | 404 (95.51%) | **0.001** |
| | Yes | 63 (2.31%) | 44 (1.91%) | 19 (4.49%) | |
| Hypertension | No | 1960 (71.95%) | 1679 (72.97%) | 281 (66.43%) | **0.006** |
| | Yes | 764 (28.05%) | 622 (27.03%) | 142 (33.57%) | |
| Chronic liver disease | No | 2616 (96.04%) | 2214 (96.22%) | 402 (95.04%) | 0.252 |
| | Yes | 108 (3.96%) | 87 (3.78%) | 21 (4.96%) | |
| Liver Cirrhosis | No | 2652 (97.36%) | 2246 (97.61%) | 406 (95.98%) | 0.055 |
| | Yes | 72 (2.64%) | 55 (2.39%) | 17 (4.02%) | |
| Asthma | No | 24 (0.88%) | 18 (0.78%) | 6 (1.42%) | |
| | Yes | 2700 (99.12%) | 2283 (99.22%) | 417 (98.58%) | |
| Coronary artery disease | No | 2511 (92.18%) | 2141 (93.05%) | 370 (87.47%) | <**0.001** |
| | Yes | 213 (7.82%) | 160 (6.95%) | 53 (12.53%) | |
| Chronic Renal Insufficiency | No | 2631 (96.59%) | 2238 (97.26%) | 393 (92.91%) | <**0.001** |
| | Yes | 93 (3.41%) | 63 (2.74%) | 30 (7.09%) | |
| Heart Failure | No | 2720 (99.85%) | 2297 (99.83%) | 423 (100.00%) | 1.00 |
| | Yes | 4 (0.15%) | 4 (0.17%) | 0 (0.00%) | |
| Dyslipidemia | No | 2721 (99.89%) | 2298 (99.87%) | 423 (100.00%) | 0.457 |
| | Yes | 3 (0.11%) | 3 (0.13%) | 0 (0.00%) | |
| CT findings | Normal | 8 (0.29%) | 8 (0.35%) | 0 (0.00%) | 0.566 |
| | Consolidation | 68 (2.50%) | 55 (2.39%) | 13 (3.07%) | |
| | Crazy paving | 28 (1.03%) | 23 (1.00%) | 5 (1.18%) | |
| | Ground Glass opacity | 1202 (44.13%) | 1033 (44.89%) | 169 (39.95%) | |
| | Others not related to COVID-19 | 2 (0.07%) | 2 (0.09%) | (0.00%) | |

±25.04, p<0.001). The Glasgow coma scale was significantly (p<0.001) lower in the critical group (14.10±2.33) than the non-critical group (14.80±0.78), Table 2.

## 3.6. Laboratory investigations

Critically ill patients were associated with higher serum levels of CRP (p<0.001), Serum ferritin (p = 0.003), TLC (p<0.001), INR (p<0.001), serum potassium (p = 0.023), HCO3 (p = 0.029), and Moreover, total bilirubin (p<0.001), direct bilirubin (p<0.001), serum albumin (p = 0.001), ALT (p = 0.008), AST (p<0.001), and serum creatinine (p = 0.007) were higher in critically ill patients, Table 2.

**Table 2. Clinical and Laboratory characteristics of the included patients.**

| Parameters | Number of population | Non-critically COVID-19 | Critically COVID-19 | P-value |
|---|---|---|---|---|
| SBP | 2,004 | 120.67±14.07 | 120.66±18.25 | 0.762 |
| DBP | 859 | 77.80±9.84 | 76.87±12.30 | 0.577 |
| Pulse | 2,492 | 89.75±13.90 | 92.61±16.72 | **0.006** |
| Temperature | 2,162 | 37.60±0.88 | 37.84±0.81 | **<0.001** |
| Respiratory rate | 1,809 | 22.91±8.19 | 27.01±14.77 | **<0.001** |
| Oxygen Saturation | 2,454 | 91.04±6.23 | 85.73±10.6 | **<0.001** |
| Glasgow coma scale | 2,724 | 14.80±0.78 | 14.10±2.33 | **<0.001** |
| FiO2 | 123 | 20.71±28.08 | 35.5±33.91 | **0.004** |
| Hemoglobin (gm/dl) | 1,319 | 12.45±7.20 | 13.13±9.96 | 0.845 |
| Platelet count (*1000/cmm) | 1,306 | 252.02±116.06 | 269.13±132.79 | 0.054 |
| TLC (*1000/cmm) | 1,628 | 10.74±19.92 | 14.21±27.73 | **<0.001** |
| INR | 1,536 | 0.65±0.75 | 0.77±0.65 | **<0.001** |
| PTT | 550 | 37.25±22.12 | 46.56±56.80 | 0.631 |
| Serum creatinine (mg/dl) | 1,539 | 1.67±2.76 | 2.24±3.12 | **0.007** |
| Serum sodium (mEq/L) | 856 | 136.37±7.70 | 137.83±9.32 | 0.157 |
| Serum potassium (mEq/L) | 1,319 | 4.25±2.65 | 4.70±4.32 | **0.023** |
| PH | 1,134 | 7.34±0.42 | 7.36±0.24 | 0.787 |
| PaCO2 mmHg | 770 | 34.07±9.58 | 36.17±12.82 | 0.113 |
| HCO3 mEq/L | 1,249 | 23.12±11.39 | 27.62±31.04 | **0.029** |
| PaO2 mmHg | 729 | 64.43±26.09 | 55.22±25.04 | **<0.001** |
| Total bilirubin (mg/dl) | 1,155 | 0.30±0.55 | 0.49±0.75 | **<0.001** |
| Direct bilirubin (mg/dl) | 1,119 | 0.13±0.31 | 0.24±0.50 | **<0.001** |
| Serum albumin (g/dl) | 1,182 | 3.77±0.69 | 3.50±0.72 | **0.001** |
| ALT (U/L) | 620 | 46.55±43.08 | 60.15±56.26 | **0.008** |
| AST (U/L) | 586 | 55.62±55.45 | 75.62±100.25 | **<0.001** |
| CRP (mg/L) | 710 | 52.19±77.79 | 88.17±93.35 | **<0.001** |
| ESR (first hour) | 375 | 99.33±80.34 | 79.67±88.16 | **0.032** |
| Serum ferritin (ng/ml) | 599 | 683.38±436.78 | 1039.59±714.03 | **0.003** |
| D-dimer (microgm/ml) | 336 | 1108.5±756.1 | 1461.1±1059.3 | 0.053 |
| Fibrinogen (mg/dl) *100 if presented in g/L | 216 | 76.50±73.08 | 98.26±80.20 | 0.135 |

## 3.7. Univariate logistic regression

Our analysis demonstrated a significant association between many variables and critical illness. In terms of demographic variables age >60 years and history of contact were observed to increase the risk of critically illness (OR = 1.29, 95% CI [1.04, 1.59], p = 0.016) and (OR = 1.47, 95% CI [1.15, 1.88], p = 0.002), respectively. Regarding the COVID-19 symptoms, critically illness was significantly associated with hemoptysis (OR = 3.65, 95% CI [1.02, 12.99], p = 0.046), fatigue (OR = 1.52, 95% CI [1.21, 1.91], p<0.001), anorexia (OR = 1.88, 95% CI [1.16, 3.06], p = 0.010), arthralgia (OR = 2.29, 95% CI [1.66, 3.17], p<0.001), and myalgia (OR = 2.09, 95% CI [1.55, 2.82], p<0.001). In terms of co-morbidities, patients with diabetes (OR = 1.71, 95% CI [1.38, 2.12], p<0.001), cancer (OR = 2.41, 95% CI [1.39, 4.17], p = 0.002), hypertension (OR = 1.36, 95% CI [1.09, 1.70], p = 0.006), and coronary artery diseases (OR = 1.91, 95% CI [1.37, 2.66], p<0.001) were associated with higher risk of critical illness, Table 3.

**Table 3. Univariate regression of risk factors.**

| Variable | OR (CI) | p-value |
|---|---|---|
| Age >60 years | 1.29 (1.04, 1.59) | **0.016** |
| History of contact | 1.47 (1.15, 1.88) | **0.002** |
| Hemoptysis | 3.65 (1.02, 12.99) | **0.046** |
| Fatigue | 1.52 (1.21, 1.91) | **<0.001** |
| Anorexia | 1.88 (1.16, 3.06) | **0.010** |
| Arthralgia | 2.29 (1.66, 3.17) | **<0.001** |
| Myalgia | 2.09 (1.55, 2.82) | **<0.001** |
| Diabetes | 1.71 (1.38, 2.12) | **<0.001** |
| Cancer | 2.41 (1.39, 4.17) | **0.002** |
| Hypertension | 1.36 (1.09, 1.70) | **0.006** |
| Coronary artery disease | 1.91 (1.37, 2.66) | **<0.001** |
| Chronic Renal Insufficiency | 2.71 (1.73, 4.24) | **<0.001** |
| NLR | 3.78 (2.78, 5.13) | **<0.001** |
| Pulse | 1.013 (1.006, 1.02) | **<0.001** |
| Temperature | 1.42 (1.23, 1.65) | **<0.001** |
| Respiratory rate | 1.03 (1.02, 1.04) | **<0.001** |
| Oxygen Saturation | 0.92 (0.90, 0.94) | **<0.001** |
| Glasgow coma scale | 0.68 (0.63, 0.75) | **<0.001** |
| FiO2 | 1.015 (1.002, 1.028) | **0.019** |
| Platelet count (*1000/cmm) | 1.001 (1.00, 1.002) | **0.046** |
| INR | 1.23 (1.04, 1.45) | **0.012** |
| Serum creatinine (mg/dl) | 1.060 (1.01, 1.113) | **0.019** |
| Serum sodium (mEq/L) | 1.02 (1.001, 1.042) | **0.041** |
| PaCO2 mmHg | 1.02 (1.003, 1.035) | **0.019** |
| HCO3 mEq/L | 1.012 (1.001, 1.023) | **0.032** |
| PaO2 mmHg | 0.989 (0.982, 0.996) | **0.003** |
| Total bilirubin (mg/dl) | 1.53 (1.23, 1.89) | **<0.001** |
| Direct bilirubin (mg/dl) | 2.11 (1.46, 3.05) | **<0.001** |
| Serum albumin (g/dl) | 0.58 (0.43, 0.81) | **0.001** |
| ALT (U/L) | 1.005 (1.002, 1.009) | **0.005** |
| AST (U/L) | 1.004 (1.001, 1.006) | **0.009** |
| CRP (mg/L) | 1.005 (1.002, 1.008) | **0.002** |
| ESR (first hour) | 0.997 (0.994, 1.00) | **0.044** |
| Serum ferritin (ng/ml) | 1.001 (1.001, 1.002) | **<0.001** |
| D-dimer (microgm/ml) | 1.00 (1.00, 1.001) | **0.011** |

## 3.8. Predictors of critical illness

Multivariate analysis was performed based on four models; 1) demographic and clinical symptoms; 2) vital signs; 3) Co-morbidities; and 4) laboratory investigations. Model 1 showed that age >60 years (OR = 1.30, 95% CI [1.05, 1.61], p = 0.014), fatigue (OR = 1.30, 95% CI [1.02, 1.65], p = 0.029), arthralgia (OR = 1.75, 95% CI [1.20, 2.54], p = 0.003), and myalgia (OR = 1.51, 95% CI [1.08, 2.11], p = 0.015) can be considered as independent predictors for COVID-19 critical illness. Model 2 demonstrated that high pulse (OR = 1.01, 95% CI [1.009, 1.03], p<0.001), temperature (OR = 1.25, 95% CI [1.04, 1.48], p = 0.012), low oxygen saturation (OR = 0.93, 95% CI [0.91, 0.95], p<0.001), and low Glasgow coma scale (OR = 0.75, 95% CI [0.67, 0.84], p<0.001) are significant predictors for COVID-19 critical illness. According to

model 3, critical COVID-19 can be predicted based on the presence of the following co-morbidities: Diabetes (OR = 1.62, 95% CI [1.26, 2.08], p<0.001), cancer (OR = 2.47, 95% CI [1.41, 4.35], p = 0.002), coronary artery diseases (OR = 1.56, 95% CI [1.09, 2.22], p = 0.014), and chronic renal insufficiency (OR = 2.81, 95% CI [1.79, 4.43], p<0.001). Regarding model 4, only ALT (OR = 1.12, 95% CI [1.0092, 1.24], p = 0.033), AST (OR = 1.13, 95% CI [1.00, 1.28], p = 0.036), and Serum ferritin (OR = 1.004, 95% CI [1.0003, 1.008], p = 0.031) can be used as predictors for COVID-19 critical illness, Table 4.

**Table 4. Multivariate analysis of risk factors.**

| Models | | Multivariate analysis | P-value | Hosmer-Lemeshow goodness-of-fit test | Chi$^2$ |
| --- | --- | --- | --- | --- | --- |
| | | | | | p-value |
| Model 1* | Age >60 years | 1.30 (1.05, 1.61) | **0.014** | 2.98 | **0.5618** |
| | History of contact | 1.20 (0.93, 1.56) | 0.151 | | |
| | Hemoptysis | 3.14 (0.82, 11.98) | 0.094 | | |
| | Fatigue | 1.30 (1.02, 1.65) | **0.029** | | |
| | Anorexia | 1.04 (0.60, 1.78) | 0.886 | | |
| | Arthralgia | 1.75 (1.20, 2.54) | **0.003** | | |
| | Myalgia | 1.51 (1.08, 2.11) | **0.015** | | |
| Model 2** | Pulse | 1.01 (1.009, 1.03) | **<0.001** | 21.38 | 0.0062 |
| | Temperature | 1.25 (1.04, 1.48) | **0.012** | | |
| | Respiratory rate | 1.01 (0.99, 1.03) | 0.181 | | |
| | Oxygen Saturation | 0.93 (0.91, 0.95) | **<0.001** | | |
| | Glasgow coma scale | 0.75 (0.67, 0.84) | **<0.001** | | |
| Model 3*** | Diabetes | 1.62 (1.26, 2.08) | **<0.001** | | |
| | Cancer | 2.47 (1.41, 4.35) | **0.002** | | |
| | Hypertension | 0.93 (0.71, 1.22) | 0.636 | | |
| | Coronary artery disease | 1.56 (1.09, 2.22) | **0.014** | | |
| | Chronic Renal Insufficiency | 2.81 (1.79, 4.43) | **<0.001** | | |
| Model 4**** | NLR | 0.87 (0.12, 6.18) | 0.892 | 4.05 | **0.8526** |
| | FiO2 | 1.01 (0.98, 1.05) | 0.246 | | |
| | Platelet count (*1000/cmm) | 0.997 (0.991, 1.00) | 0.460 | | |
| | INR | 0.31 (0.009, 10.11) | 0.516 | | |
| | Serum creatinine (mg/dl) | 1.11 (0.70, 1.75) | 0.648 | | |
| | Serum sodium (mEq/L) | 1.02 (0.88, 1.18) | 0.732 | | |
| | PaCO2 mmHg | 1.12 (0.96, 1.31) | 0.144 | | |
| | HCO3 mEq/L | 0.96 (0.58, 1.59) | 0.885 | | |
| | PaO2 mmHg | 0.91 (0.81, 1.03) | 0.173 | | |
| | Total bilirubin (mg/dl) | 1.35 (0.10, 17.8) | 0.816 | | |
| | Direct bilirubin (mg/dl) | 14.25 (0.37, 547.5) | 0.153 | | |
| | Serum albumin (g/dl) | 2.11 (0.66, 6.72) | 0.205 | | |
| | ALT (U/L) | 1.12 (ss) | **0.033** | | |
| | AST (U/L) | 1.13 (1.00, 1.28) | **0.036** | | |
| | CRP (mg/L) | 0.98 (0.96, 1.00) | 0.122 | | |
| | ESR (first hour) | 0.99 (0.98, 1.01) | 0.824 | | |
| | Serum ferritin (ng/ml) | 1.004 (1.0003, 1.008) | **0.031** | | |
| | D-dimer (microgm/ml) | 0.998 (0.996, 1.0008) | 0.236 | | |

* Adjusted model to demographic variables.

** Adjusted model to Vital signs.

*** Adjusted model to Clinical characteristics.

****Adjusted model to laboratory investigations.

### 3.9. Gender disparity amongst the studied population

Male patients were more likely to be smokers (p <0.001), had a history of close contact with positive cases (p = 0.043), being asymptomatic (p = 0.043), S1 Table.

## 4. Discussion

The landscape of the COVID-19 pandemic has changed over the past few months. The number of confirmed cases in the EMR rose sharply to more than 1,534,000 cases by the end of July 2020, with about 40,000 deaths reported, resulting in a 2.6% mortality rate [6]. Growing evidence shows that the different distributions of epidemiological characteristics between the affected countries may influence mortality rates. In addition, the clinical spectrum of COVID-19 disease appears to be broad, ranging from asymptomatic infection, mild upper respiratory tract disease, respiratory failure to severe viral pneumonia or death. The assessment of risk factors and predictors of the severity of the disease and the possibility of death is a very important issue for the prediction of the possible outcome.

In this study, we highlighted many risk factors that can be used as independent predictors for the severity of COVID-19. Age, especially patients older than 60 years, were significantly associated with worse outcomes of COVID-19. It has been observed that older age groups are more susceptible to infection and severe presentation, with higher mortality rates than younger patients, since the initial COVID-19 case clusters were reported [29]. While it was initially unclear whether the difference between age groups reflects the lower risk of severe disease or lower susceptibility to infection, recent reports demonstrated that the young population had similar susceptibility to older age groups; which, in return, highlights that the difference between age groups stems from the ability to fight the infection [30]. Several studies proposed many pathogenic mechanisms behind the critical illness in the elderly population, including low levels of angiotensin-converting enzyme 2 (ACE2) in the elderly [31], age-dependent difficulty in removing particles from small airways [32], excessive release of inflammatory mediators in elderly "inflammaging" [33], incompetent immune response, and high frequency of comorbidities in the elderly population [29]. Recent retrospective studies from Egypt [25], Iran [34], and Pakistan [35] demonstrated that older age groups were more susceptible to severe disease and death from COVID-19. This was in line with other reports from outside of EMR countries [36, 37].

Recent surveillance had shown that male patients were more susceptible to serious infection and mortality with COVID-19 [38], and this gender-specific difference had been observed in previous outbreaks of SARS [39]. In contrast, our study showed an insignificant difference between both genders in terms of COVID-19 severity. This observation was different from what was reported in the other reports from EMR countries [25, 34, 35]. This can be explained as we here investigate the risk of severity, not mortality. Similarly, Nasiri et al. showed that there was no significant difference between males and females in terms of ICU admissions [40]. Another Egyptian study of 260 patients with COVID-19 showed that there was no significant association between male gender and the risk of critical illness (OR = 1.84, 95% CI [0.80, 4.48], p = 0.156) [41]. In Italy, Ciceri et al. [42] reported that there was no significant association between female gender and risk of critical illness (OR = 1.09, 95% CI [0.69, 1.72], p = 0.70).

It is well documented that comorbidities in COVID-19 patients are major contributors to serious presentation and death. For example, poorly controlled diabetes was found to significantly increase the risk of composite adverse outcomes, including death, among COVID-19 cases [43]. Likewise, malignancies and chronic renal diseases were reported to be significant predictors of COVID-19 critical illness and mortality [44]. The preexistence of cardiovascular

or chronic lung diseases worsens the outcomes of COVID-19 as well [45, 46]. In our study, diabetes mellitus, cardiovascular diseases, chronic renal disease, and cancer were significantly associated with a higher risk of critical illness. In a large cohort from China, hypertension, diabetes, and chronic obstructive pulmonary diseases were prevalent among severe COVID-19 cases [47]. Patients with preexistent comorbidities, such as diabetes and cardiovascular diseases, were more likely to suffer from the severe disease in another report of COVID-19 cases [48].

Low Oxygen saturation is one of the most important clinical features of COVID-19 patients. Our study showed that lower oxygens saturation was significantly associated with an increased risk of critical illness. In agreement, Petrilli et al. [36] showed that patients with deficient oxygen saturation (less than <88%) were associated with increased risk of critical COVID-19 (OR = 3.67, 95% CI [2.78, 4.8], p<0.001). Also, we found that patients with elevated temperature and increased pulsation were associated with an increased risk of critical illness. In contrast, Petrilli et al. could not find any significant association between elevated temperature and critical illness [36].

In our study, the critically ill patients showed significant leukocytosis. Similarly Zhao et al found that elderly patients with co-morbidities showed significant leukocytosis [49]. Patients with COVID-19 have a blood picture characterized by normal or low WBC count and decreased lymphocyte level [50]. Similarly, the elevation of CRP levels was observed in the majority of COVID-19 patients. Moreover, high levels of CRP and inflammatory biomarkers are considered one of the main characteristics of COVID-19 patients [51]. It was reported that early elevation of CRP has the strongest association with mechanical ventilation or mortality [52]. Furthermore, liver changes were one of the most common findings in the COVID-19 infected subjects. Liver injury may happen in COVID-19. It had been reported that that 14.8–53.1% of COVID-19 patients had elevated liver enzymes and bilirubin. Moreover, the degree of liver damage is proportional to the severity of COVID-19 [53, 54]. In patients with COVID-19, the serum level of ferritin was significantly elevated in critically ill patients. Similarly, it had been reported that serum level of ferritin increase with case deterioration and was significantly elevated in non-survivors compared with survivors [37]. Shoenfeld reported that the ferritin H-chain played a role in stimulating macrophages to increase inflammatory cytokine secretion. Thus, it became apparent to understand the pathogenesis of hyperferritinemia syndrome, including COVID-19 infection [55]. Our findings showed that ALT, AST, and serum ferritin could be used as a significant predictor for COVID-19 severity. Interestingly, our findings demonstrated that there was no significant association between the level of D-dimer and the severity of COVID-19. This finding was also observed by Ramadan et al. in another Egyptian study [41].

This article is one of the first reports about the predictors of COVID-19 severity in Egypt. However, we acknowledge that the present study had some limitations. First, data of the present study were collected retrospectively with some potential biases, such as case ascertainment bias and misclassification. Second, we might have overestimated the importance of chronic disease at risk of hospital admission due to the substantial heterogeneity among the included patients. Third, our patients were from a single geographical area, treated within a single health system; so factors correlated with poor outcomes might differ elsewhere. The lack of medical history at baseline was an additional limitation, recent reports highlighted that some medications, such as angiotensin-converting enzyme (ACE) inhibitors and angiotensin II receptor blockers (ARBs), may facilitate severe infection [56]. Besides, a high frequency of obesity among patients admitted with sever COVID-19 [57]; however, there were no available data concerning obesity in the present study.

## 5. Conclusion

In conclusion, COVID-19 is a growing pandemic with an unprecedented spread rate and profound impact on the health of specific subsets of affected patients. In the present report, we demonstrated that many factors are associated with COVID-19 critical illness, including older age groups, fatigue, elevated temperature, increased pulse, lower oxygen saturation, the preexistence of diabetes, malignancies, cardiovascular disease, renal diseases, and pulmonary disease. Moreover, elevated serum levels of ALT, AST, and ferritin are associated with worse outcomes. Further studies are required to identify independent predictors of mortality for patients with COVID-19.

## Supporting information

**S1 Table. The difference in clinical and laboratory characteristics of the included patients according to gender.**
(DOCX)

## Author Contributions

**Conceptualization:** Dalia Omran, Hend Shousha, Muhammad Abdel Ghaffar.

**Data curation:** Mohamed Alsheikh, Enass El Sayed, Shimaa Afify, Samah Abdel Hafez, Khaled Elkelany, Ayman Eltayar, Omnia Ali, Lamiaa Kamal, Ahmed Heiba.

**Formal analysis:** Eshak Bahbah.

**Methodology:** Dalia Omran.

**Supervision:** Dalia Omran, Mohamed Al Soda, Muhammad Abdel Ghaffar.

**Writing – original draft:** Eshak Bahbah, Ahmed Elgebaly.

**Writing – review & editing:** Dalia Omran, Gamal Esmat, Muhammad Abdel Ghaffar.

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
