## [Decision Letter · Decision Letter 0]

12 Feb 2021

PONE-D-21-00016

Predictors of severity and development of critical illness in Egyptian COVID-19 patients: A multicenter study

PLOS ONE

Dear Dr. Omran,

Thank you for submitting your manuscript to PLOS ONE. After careful consideration, we feel that it has merit but does not fully meet PLOS ONE’s publication criteria as it currently stands. Therefore, we invite you to submit a revised version of the manuscript that addresses the points raised during the review process and particularly provides the additional data requested.

We look forward to receiving your revised manuscript.

Kind regards,

Andreas Zirlik, MD

Academic Editor

PLOS ONE

Journal Requirements:

Reviewers' comments:

Reviewer's Responses to Questions

**Comments to the Author**

1. Is the manuscript technically sound, and do the data support the conclusions?

Reviewer #1: Yes

Reviewer #2: Yes

2. Has the statistical analysis been performed appropriately and rigorously? 

Reviewer #1: Yes

Reviewer #2: I Don't Know

3. Have the authors made all data underlying the findings in their manuscript fully available?

Reviewer #1: Yes

Reviewer #2: No

4. Is the manuscript presented in an intelligible fashion and written in standard English?

Reviewer #1: No

Reviewer #2: Yes

5. Review Comments to the Author

Reviewer #1: The retrospective and multicenter analysis by Omran et al. summs up the total number of hospitalized patients in the GOTHI affiliated hospitals with positive COVID-19 PCR from March to July 2020.

Out of 2724 patients 423 were rated critically ill.

Several factors could be identified which predicted worse outcome (to be rated critically ill).

This is a large study with little selection bias as all COVID-19 patients of the hospitals analysed were included. PCR as the gold standard to detect SARS-CoV2 was used. The study was approved by the respective ethics committee. The data confirms many aspects already described in other populations within an egypt population. Some factors seem to diverge (such as lacking increased risk in male patients) and are discussed accordingly.

There are some aspects to be clarified or discussed in more detail:

1.) Please state how many COVID-19 tests were performed in the population analyzed and what the positive-rate of these tests was.

2.) Have there been specific algorithms to trigger testing and was PCR the only test? Or did PCR confirm antigen tests for example? Did all positively tested patients show COVID-related symptoms or does the study also include "chance findings" in asymptomatic patients tested for other reasons (potential contact person...) How many patients were rated ill or critically ill due to COVID-19 and how many due to other diseases?

3.) What was the rationale for selecting the analysed time frame from March to July?

4.) The authors state a fatality rate of 5.8 for egypt (citation). How was the rate in these hospitalized patients?

5.) With respect to comorbidities some numbers seem to be unusual. There are only 9 COPD Patients in the whole cohort despite more than 10% smokers. And only 1 out of 2301 patients in the non-critical group was diagnosed with arrhythmias. Which arrhythmias were analysed as atrial fibrillation is obviously not included. I suggest to only display clinical characteristics for comorbidities with reliable data (which of course cannot always be available in retrospective analysis)

6.) FiO2 is stated to be 20.71+-28.08 in the non critical group. Assuming 21% oxygen within natural air, this seems to be to low as some patients obviously received oxygen. How was oxygen calculated into FiO2 if most patients were not on respirators?

7.) Regarding the unchanged risk between the two genders; is it possible that other risk factors might be differing, too. Are, for instance, female patients older in this cohort?

Reviewer #2: The manuscript entitled "Predictors of severity and development of critical illness in Egyptian COVID-19 patients: A multicenter study" by Omran et al. gives an overview on the factors leading to a severe illness during the "first wave" of the COVID19 pandemic. The conclusion of the authors is that higher age (> 60 years), preexisting comorbidities, more symptoms presenting at baseline (I assume the data are baseline data), and unfavorable vital signs (high heart rate, low oxygen saturation) are associated with a more severe COVID19 illness.

The manuscript is well written and most data used in their statistical analysis are given. Although the results are not surprising, regional analysis of the COVID19 disease is of interest.

However, there are some points that need to be discussed and further described.

Major points:

1) Additional information regarding the Ethics approval is missing (registration number or something comparable).

2) Timing of the data collection (I assume these are baseline data)

3) Have all data been collected? How did the authors manage missing data (“unknown” counted as “no”; insert the number of laboratory samples that could be analyzed). This needs to be addressed and data must be given in a this transparent way

4) No gender-specific differentiation in laboratory findings has been made, this is of particular importance for the liver enzymes that are associated with a more severe illness.

5) Lack of medical history at baseline (at least substance classes like ACE-inhibitors, statins, oral antidiabetic drugs etc.) -> this should also be statistically analyzed. Although hypertension predicted a more severe illness, the blood pressure was comparable in the "critical" and "non-critical" group. Of note, ACE-inhibitors are currently discussed to influence the severity of COVID19.

6) BMI (or height & weight) are missing as demographic variables (see as reference Simonnet et al., Obesity 2020 PMID 32271993)

7) The duration of symptoms is not given (symptom onset untill positive RT-PCR)

8) No information is given on fatal outcomes. It would be reasonable to calculate the prediction value of different biomarkers and variables for a more severe illness for fatal outcome too. This would help to classify the clinical impact of the results described in the manuscript

Minor points:

- Just some minor spelling mistakes

- Consider the definition of potentially unclear preexisting comorbidities (for example in a supplementary file). How was hypertension defined? How was heart failure defined? Cut-off for chronic renal insufficiency? What was the definition for dyslipidemia? What was the cut-off value for fever (maybe split this in fever > 38.5°C and "subfebrile temperature 37 - 38.5°C"?

My summary:

It is an interesting data set. However, given its retrospective nature, a more detailled analysis must be performed before any conclusion can be drawn.

6. PLOS authors have the option to publish the peer review history of their article (what does this mean?). If published, this will include your full peer review and any attached files.

Reviewer #1: No

Reviewer #2: No

---

## [Author Response · Author response to Decision Letter 0]

7 Apr 2021

Dear Andreas Zirlik, MD, Academic Editor of PLOS ONE,

We are happy to receive the reviewers' comments, which are excellent additions to the quality of our manuscript. We have done all the changes recommended by the reviewers. Below, we are attaching a table with all changes and our point-by-point response to reviews' comments.

Reviewer #1

1. Please state how many COVID-19 tests were performed in the population analyzed and what the positive-rate of these tests was.

Authors' response: Thank you so much for your valuable insights and comments, which significantly improved the quality of our manuscript. According to the national protocol issued by the MoH of Egypt, suspected COVID-19 cases are hospitalized on the basis of positive PCR tests. Thus, our study included only positive PCR cases. Owing to the retrospective nature of the present study and the fact that the databases of participating hospitals were confined to positive cases only, it was not feasible to calculate the positive rate from the present study.

2. Have there been specific algorithms to trigger testing and was PCR the only test? Or did PCR confirm antigen tests for example? Did all positively tested patients show COVID-related symptoms or does the study also include "chance findings" in asymptomatic patients tested for other reasons (potential contact person...) How many patients were rated ill or critically ill due to COVID-19 and how many due to other diseases?.

Authors' response: Thank you so much for your comment. According to the Egyptian national guideline, PCR testing is recommended for symptomatic cases and close contacts of positive cases.

As our study retrieved the data of the confirmed COVID-19 admitted to hospitals affiliated to the GOTHI, all included patients were either symptomatic or close contacts; while there were no cases with “chance findings”.

The confirmation of the COVID-19 was based on PCR test only. The severe cases were defined as the presence of radiological evidence of more than 50% lung infiltrate plus one of the following: respiratory rate of ≥ 30 breaths per minute; oxygen saturation (SaO2) <94% while breathing ambient air at rest; or ARDS that is defined as arterial oxygen partial pressure (PaO2) to a fraction of inspired oxygen (FiO2) (PaO2: FiO2) of ≤300 mmHg. Thus, the definition of severity depends solely on COVID-19, and not any other disease.

3. What was the rationale for selecting the analyzed time frame from March to July?.

Authors' response: Thank you for your comments. The selection of time frame depended on the availability of the data within the participating hospitals and the duration during which the present study was conducted. The data collection started at July 2020, while the first positive case in Egypt was reported at the begging of March 2020. Thus, the timeframe of data collection covers the period from March to July 2020.

4. The authors state a fatality rate of 5.8 for Egypt (citation). How was the rate in these hospitalized patients. 

Authors' response: Thank you for your comments. The mortality rate among hospitalized patients with COVID-19 in Egypt was reported to be 6.7% (1), which rises to 40% in severe and critically-ill cases (2).

We are currently working on another manuscript from the same datasets, which aims to present the characteristics of COVID-19-related fatalities and the predictors of mortality.

5. With respect to comorbidities some numbers seem to be unusual. There are only 9 COPD Patients in the whole cohort despite more than 10% smokers. And only 1 out of 2301 patients in the non-critical group was diagnosed with arrhythmias. Which arrhythmias were analysed as atrial fibrillation is obviously not included. I suggest to only display clinical characteristics for comorbidities with reliable data (which of course cannot always be available in retrospective analysis).

Authors' response: Thank you for your comments. Owing to the retrospective nature of the present study, we acknowledge the possibility of misclassification and ascertainment bias. In order to account for such limitations, we displayed only the clinical characteristics for comorbidities with reliable data. Besides, we acknowledged this limitation in the discussion section.

6. FiO2 is stated to be 20.71+-28.08 in the non critical group. Assuming 21% oxygen within natural air, this seems to be to low as some patients obviously received oxygen. How was oxygen calculated into FiO2 if most patients were not on respirators?

Authors' response: Thank you for your comments. The data input at the level of one of our hospitals at the FiO2 variable was replaced with O2 requirement (L/min), and that was mistakenly bypassed at the level of data integrity revision. So, we have taken corrective actions for full data revision and resubmission accordingly.

Reviewer #2

1. Additional information regarding the Ethics approval is missing (registration number or something comparable).

Authors' response: The present study was initiated after obtaining the protocol approval from the General Organization for Teaching Hospitals and Institutes (GOTHI) responsible ethics committees in Egypt (IRB:HAM00122).

2. Timing of the data collection (I assume these are baseline data)

Authors' response: Thank you for your comments. The selection of time frame depended on the availability of the data within the participating hospitals and the duration during which the present study was conducted. The data collected started at July 2021, while the first positive case in Egypt was reported at the begging of March 2020. The present data were collected at the first day of hospitalization of each patient.

3. Have all data been collected? How did the authors manage missing data (“unknown” counted as “no”; insert the number of laboratory samples that could be analyzed). This needs to be addressed and data must be given in a this transparent way

Authors' response: Thank you for your comments. The missed data were left blank. We presented the number of available population at each variable to ensure transparency.

4. No gender-specific differentiation in laboratory findings has been made, this is of particular importance for the liver enzymes that are associated with a more severe illness.

Authors' response: Thank you for your comments. We have conducted a subgroup analysis to found the existence of significant differences between both genders. We found that male patients were more likely to be smokers and asymptomatic. While female patients had higher values for liver enzymes and laboratory markers. The subgroup analysis is present in supplementary table 1.

5. Lack of medical history at baseline (at least substance classes like ACE-inhibitors, statins, oral antidiabetic drugs etc.) -> this should also be statistically analyzed. Although hypertension predicted a more severe illness, the blood pressure was comparable in the "critical" and "non-critical" group. Of note, ACE-inhibitors are currently discussed to influence the severity of COVID19.

Authors' response: Thank you for your comments. While we agree with on the importance of investigating the impact of medical history, it was not feasible to retrieve such data from the hospitals’ databases unfortunately. We have discussed this point in the limitation.

6. BMI (or height & weight) are missing as demographic variables (see as reference Simonnet et al., Obesity 2020 PMID 32271993)

Authors' response: Thank you for your comments. While we agree with on the importance of investigating the impact of obesity, it was not feasible to retrieve such data from the hospitals’ databases unfortunately. We have discussed this point in the limitation.

7. The duration of symptoms is not given (symptom onset until positive RT-PCR)

Authors' response: Thank you for your comments. It was not feasible to retrieve such data from the hospitals’ databases unfortunately. We have discussed this point in the limitation.

8. No information is given on fatal outcomes. It would be reasonable to calculate the prediction value of different biomarkers and variables for a more severe illness for fatal outcome too. This would help to classify the clinical impact of the results described in the manuscript.

Authors' response: Thank you for your comments. We are currently working on another manuscript from the same datasets, which aims to present the characteristics of COVID-19-related fatalities and the predictors of mortality.

9. Just some minor spelling mistakes

Authors' response: Thank you for your comments. The English language was revised thoroughly.

10. Consider the definition of potentially unclear preexisting comorbidities (for example in a supplementary file). How was hypertension defined? How was heart failure defined? Cut-off for chronic renal insufficiency? What was the definition for dyslipidemia? What was the cut-off value for fever (maybe split this in fever > 38.5°C and "subfebrile temperature 37 - 38.5°C"?

Authors' response: Thank you for your comments. We have added the definition of each comorbid condition in the methodology section.

References:

1. Ghweil AA, Hassan MH, Mohamed AK, Mohamed AO, Mohammed HM, Abdelazez AA, et al. 

Characteristics, Outcomes and Indicators of Severity for COVID-19 Among Sample of ESNA Quarantine Hospital’s Patients, Egypt: A Retrospective Study

. Infect Drug Resist [Internet]. 2020 Jul 17 [cited 2020 Aug 4];Volume 13:2375–83. Available from: https://www.dovepress.com/characteristics-outcomes-and-indicators-of-severity-for-covid-19-among-peer-reviewed-article-IDR

2. Ramadan HKA, Mahmoud MA, Zakaria M, Aburahma, Elkhawaga AA, El-Mokhtar MA, et al. Predictors of severity and co-infection resistance profile in COVID-19 patients: First report from upper Egypt. Infect Drug Resist [Internet]. 2020 [cited 2021 Mar 28];13:3409–22. Available from: /pmc/articles/PMC7547142/

---

## [Decision Letter · Decision Letter 1]

3 Aug 2021

Predictors of severity and development of critical illness in Egyptian COVID-19 patients: A multicenter study

PONE-D-21-00016R1

Dear Dr. Omran,

We’re pleased to inform you that your manuscript has been judged scientifically suitable for publication and will be formally accepted for publication once it meets all outstanding technical requirements.

Kind regards,

Andreas Zirlik, MD

Academic Editor

PLOS ONE

Additional Editor Comments (optional):

Reviewers' comments:

Reviewer's Responses to Questions

**Comments to the Author**

1. If the authors have adequately addressed your comments raised in a previous round of review and you feel that this manuscript is now acceptable for publication, you may indicate that here to bypass the “Comments to the Author” section, enter your conflict of interest statement in the “Confidential to Editor” section, and submit your "Accept" recommendation.

Reviewer #1: All comments have been addressed

Reviewer #2: (No Response)

2. Is the manuscript technically sound, and do the data support the conclusions?

Reviewer #1: Yes

Reviewer #2: Partly

3. Has the statistical analysis been performed appropriately and rigorously? 

Reviewer #1: Yes

Reviewer #2: Yes

4. Have the authors made all data underlying the findings in their manuscript fully available?

Reviewer #1: Yes

Reviewer #2: No

5. Is the manuscript presented in an intelligible fashion and written in standard English?

Reviewer #1: Yes

Reviewer #2: Yes

6. Review Comments to the Author

Reviewer #1: (No Response)

Reviewer #2: The authors improved the manuscript by adding information regarding transparency and data availability. Unfortunately, most major points of my previous review could not be addressed in an adequate way (mostly due to lack of data, see comments 5 – 7 of the previous review). These facts are now being discussed in the manuscript. Despite noting in their response that comorbidities have been defined in the “Methods” section, none such definitions can be found (see minor comment of the previous review).

A serious issue is the accuracy of the database of this retrospective analysis. Many mistakes have already been identified in the database after the first revision. Some statistical calculations use the number of symptoms / comorbidities given in table 1, that do not differentiate between “unknown” and “no” (see comment 3 of the previous review). The low number of sampled biomarkers (e.g., only 75 % of all patients have a documented systolic blood pressure and only 90 % of all patients have a documented oxygen saturation � table 2) implies that the number of documented diagnoses / symptoms / comorbidities (table 1) is severely underestimated. Therefore, a serious conclusion cannot be drawn. This limitation is also not addressed adequately in the discussion but reported as the conclusion.

Another serious point is to hold back information on fatal outcomes (and including these into statistical analysis) to publish these data in a different manuscript. This issue weakens the current manuscript and the proposed predictors in a significant way.

Despite the need to understand the COVID-19 disease to overcome the pandemic (and here, regional aspects must be included), the scientific impact of the current manuscript stays low. This is mostly due to the retrospective nature of the analysis and the incomplete dataset. Therefore, the manuscript should not be published in its current way. Probably the best accuracy of the data can be found in the critical ill patients and an extensive workup of these data combined with outcomes may serve this purpose.

7. PLOS authors have the option to publish the peer review history of their article (what does this mean?). If published, this will include your full peer review and any attached files.

Reviewer #1: No

Reviewer #2: No

---

## [Editor Report · Acceptance letter]

16 Sep 2021

PONE-D-21-00016R1 

Predictors of severity and development of critical illness of Egyptian COVID-19 patients: A multicenter study

Dear Dr. Omran:

I'm pleased to inform you that your manuscript has been deemed suitable for publication in PLOS ONE. Congratulations! Your manuscript is now with our production department. 

Kind regards, 

on behalf of

Univ. Prof. Dr. Andreas Zirlik 

Academic Editor

PLOS ONE